# Availability, prices and affordability of essential medicines: A cross-sectional survey in Hanam province, Vietnam

**Huong Thi Thanh Nguyen[1], Dai Xuan Dinh[1]\*, Trung Duc Nguyen[2], Van Minh Nguyen[3]**

**1** Department of Pharmaceutical Management and PharmacoEconomics, Hanoi University of Pharmacy, Hanoi city, Vietnam, **2** Pharmacy Department, 108 Military Center Hospital, Hanoi city, Vietnam, **3** Center for Population Health Sciences, Hanoi University of Public Health, Hanoi city, Vietnam

\* dinhxuandai.224@gmail.com

## Abstract

### Objective

To measure medicines' prices, availability, and affordability in Hanam, Vietnam.

### Methods

The standardized methodology developed by the World Health Organization (WHO) and Health Action International was used to survey 30 essential medicines (EMs) in 30 public health facilities and 35 private medicine outlets in 2020. The availability of medicine was computed as the percentage of health facilities in which this medicine was found on the data-collection day. International reference prices (IRPs) from Management Sciences for Health (2015) were used to compute Median Price Ratio (MPR). The affordability of treatments for common diseases was computed as the number of days' wages of the lowest-paid unskilled government worker needed to purchase medicines prescribed at a standard dose. Statistic analysis was done using R software version 4.1.1.

### Results

The mean availability of originator brands (OBs) and lowest-priced generics (LPGs) was 0.7%, 63.2% in the public sector, and 13.7%, 47.9% in the private sector, respectively. In private medicine outlets, the mean availability of both OBs and LPGs in urban areas was significantly higher than that in rural areas (p = 0.0013 and 0.0306, respectively). In the public sector, LPGs' prices were nearly equal to their IRPs (median MPRs = 0.95). In the private medicine outlets, OBs were generally sold at 6.24 times their IRPs while this figure for LPGs was 1.65. The affordability of LPGs in both sectors was good for all conditions, with standard treatments costing a day's wage or less.

### Conclusion

In both sectors, generic medicines were the predominant product type available. The availability of EMs was fairly high but still lower than WHO's benchmark. A national-scale study

**Data Availability Statement:** All relevant data are within the manuscript and its Supporting information files.

**Funding:** The author(s) received no specific funding for this work.

**Competing interests:** The authors have declared that no competing interests exist.

should be conducted to provide a comprehensive picture of the availability, prices, and affordability of EMs, thereby helping the government to identify the urgent priorities and improving access to EMs in Vietnam.

## Introduction

Essential medicines (EMs) are those that satisfy the priority healthcare requirements of the population; need to be available in a functioning health system at all times, in appropriate dosage forms, of assured quality, and at prices that individuals and the community can afford [1]. Access to safe, effective, and high-quality medicines for all people is one of the targets of the sustainable development goals. However, it is estimated that nearly two billion people have no access to basic medicines [2]. In 2019, roughly 5.2 million children under five years (2.4 million newborns) died because of reasons which can be prevented or treated through simple and affordable interventions such as adequate nutrition, immunization, and appropriate treatment of common childhood illnesses [3, 4]. Low availability, high medicine prices, and low affordability are several major barriers hindering access to EMs [2, 5].

In 1977, the first Model List of EMs was published by the World Health Organization (WHO). This document has been revised every two years. The current versions were updated in 2019 (including the 21st WHO EMs List and the 7th WHO Model List of EMs for Children) [6]. WHO has also launched a new easy-to-access, digital version of its Model List of EMs. More than 150 countries used the WHO Model Lists of EMs to compile their national essential medicines lists (NEMLs) [7]. A standard methodology for measuring medicine prices, availability, affordability, and price components was developed by the WHO and Health Action International (HAI) [8]. Hitherto, in more than 60 countries, there has been a multitude of conducted studies using this methodology [9].

Vietnam is a lower-middle-income country in Southeast Asia with a population of roughly 97.58 million inhabitants. In 2020, the national income per capita was approximately 183US$ (241US$ and 150US$ for people living in urban and rural areas, respectively) [10]. In general, the healthcare system of Vietnam can be divided into two sectors (public and private) with four administrative levels (central, provincial, district, and commune) [11]. About 77% of health professionals are working in public health facilities. The countrywide number of patient beds, doctors, and pharmacists was about 330.8 thousand beds (28.5 beds per 10,000 inhabitants), 96.2 thousand doctors (8.8 doctors per 10,000 inhabitants), and 27.5 thousand pharmacists (2.92 pharmacists per 10,000 inhabitants), respectively [12, 13]. In 2018, Vietnam possessed roughly 13,319 public health facilities, including 1,192 hospitals and 11,810 health stations. In addition, there were about 61,867 private medicine outlets stocking and trading pharmaceuticals and medical supplies all over Vietnam [13]. As per the Vietnam Ministry of Health (VMOH), in 2020, nearly 91% of the population have been covered by social health insurance. For different population groups, there are three levels of co-payment rates applied (including 0%, 5%, and 20%). For instance, the 0% rate is for people living below the breadline and children under six years of age. In other words, these people can be diagnosed and treated for diseases free of charge in the hospitals named in their health insurance cards [12].

In 2018, the VMOH promulgated the list of 1,030 modern medicines and biologicals, and 59 radiopharmaceuticals and tracers covered by health insurance with their insurance coverage ratios and payment conditions thereof [14]. In the past, these pharmaceuticals were selected based on the recommendations of healthcare professionals. In recent years, the Health Technology Assessment (HTA) has been commenced being used to generate evidence involving cost-effectiveness, thereby guaranteeing that medicines covered by health insurance were

selected based on four following important criteria: clinical effectiveness, safety, budget impact, and cost-effectiveness [12]. The majority of medicines covered by health insurance are EMs. In 1985, the first NEML was published by the VMOH. The newest version is the 7th NEML released in 2018, including 510 modern medicines and 737 herbal and traditional medicines [15]. There are several studies on medicine prices, availability, and affordability conducted in Vietnam before the year 2018 [16, 17]. From the year releasing the 7th NEML to now, in Vietnam, there has been no study conducted to survey the availability, prices, and affordability of two types of EMs: originator brands (OBs) and lowest-priced generics (LPGs). Therefore, there is an urgent need to conduct studies on EMs in Vietnam. This research was carried out to measure EMs' prices, availability, and affordability in the Hanam province of Vietnam.

## Materials and methods

This research was conducted using the standardized methodology developed by the WHO/ HAI [8, 18]. Before conducting this survey, researchers obtained approval from the ethics committee of Hanoi University of Pharmacy (reference number 10-20/PCT-HĐĐĐ). In order to facilitate data collection, researchers received an official letter of endorsement from the Hanam Department of Health. Before collecting data, data collectors gave this letter to interviewees (pharmacists). All pharmacists were informed about the objectives and methodology of this study. Verbal consent was obtained from all of the interviewees. The identity of interviewees and health facilities was kept confidential. The ethics committee did not require the research team to gain written consent from interviewees because in this study, we did not do clinical trials, did not take blood samples, and did not do any activities which could harm interviewees in any form. We only asked pharmacists about the availability and prices of medicines and therefore, it seems that the data-collection process did not have any detrimental effects on interviewees. The verbal consent procedure was approved by the ethics committee.

### Surveyed areas and health facilities

Hanam is a small province located in the Red River Delta region with an area of 861.9km$^2$ and a population of about 861.8 thousand inhabitants (1,000 people per km$^2$). The monthly average income of a citizen was approximately 175 US$ [10]. This province consists of one city and five districts. Phuly is the city of Hanam, including 12 wards. Five other areas include Thanhliem (including one town and 19 communes), Binhluc (including one town and 20 communes), Duytien (including two towns and 19 communes), Lynhan (including one town and 22 communes), and Kimbang (including one town and 18 communes) [19]. Most of the areas in this province are rural areas. The number of doctors working in health facilities was about 547 [10]. For the public sector, Hanam has roughly 137 public health facilities (with 3,350 patient beds), including 14 hospitals, 116 commune health stations, and 7 other facilities [20]. For the private sector, there are 20 pharmaceutical companies and 603 medicine outlets all over Hanam [21]. In the context of the COVID-19 outbreak, this province was selected for investigation by reason of the following rationales. The first reason was the paucity of funds and human resources. Due to the travel restrictions, the process of data collection was much easier due to the adjacency between Hanam province and the Hanoi capital. Secondly, at the time of data collection, there were no COVID-19 patients in this province and social distancing was not imposed. In addition, the friendly relation between researchers and the leaders of the Hanam Department of Health facilitated the process of data collection.

Data were collected in six areas, including Phuly city and five abovementioned districts. Areas fully met the following requirements of the WHO/HAI [8]: Each area covered a

population of about 100,000 or more; consisted of the requisite number of health facilities and being reachable within one day's travel using motorbikes from the major urban center of Hanam (Phuly city). For the public sector, six following hospitals (including Phuly Provincial General Hospital, Duytien District General Hospital, Lynhan District General Hospital, Kimbang District General Hospital, Binhluc District General Hospital, and Thanhliem District General Hospital) were selected for six surveyed areas (they are main public hospitals). In addition, in each area, four other health facilities (other hospitals, healthcare centers, or health stations) were randomly selected for the public sector. A licensed private medicine outlet in the proximity of each of the selected public health facilities was selected for the private sector. All surveyed facilities were within three hours' travel of the main public hospitals and featured on the list of health facilities and licensed medicine outlets provided by the Hanam Department of Health. Because fewer than 50% of medicines were available at five private medicine outlets, five backup facilities were visited to collect additional data. In total, there were 30 public health facilities and 35 private medicine outlets selected for this survey (S1 Table).

## Surveyed medicines

Thirty EMs selected for investigation were divided into two lists (S2 Table). The core list consisted of 14 medicines recommended for investigation by the WHO/HAI [8, 18]. The supplement list included 16 medicines chosen based on the common use in the burden of local diseases, the treatment of important national health problems, and the 2018 NEML [15]. Moreover, the suitability of selected medicines was checked through the results of a pilot test in a public health center and a private medicine outlet (as per the guidance of the WHO/HAI [8]).

## Data collection

After received standardized training, data collectors visited selected facilities in pairs and recorded whether medicines were found or not, and the prices of available medicines. A data-collection form was used to survey the availability and prices of 30 EMs in Hanam province from October to December 2020. For each medicine, data were collected for two products: OB–the original patented pharmaceutical product and LPG–the lowest-priced generic medicine in the health facility at the time of data collection. Medicines were physically observed to confirm their availability. Medicine prices were recorded from the product labels or the computers of facilities. Data collectors checked that data-collection forms were complete, accurate, and legible before leaving each facility.

## Data analysis

The computerized WHO/HAI Workbook was employed for data entry. To ensure the quality of data and the accuracy of results, data were entered twice by two people and checked using the available functions of this Workbook (including double entry and data checker).

**Availability.** The availability of medicine was calculated as the percentage of facilities in which this medicine was found on the day of data collection. R software version 4.1.1 was used for statistic analysis. The *Wilcoxon rank-sum test* was used to compare two independent groups of samples when the data were not normally distributed. For three groups or more, when the assumptions of one-way ANOVA were not met, the *Kruskal-Wallis test* and the *Dunn test* for multiple comparisons were used for comparisons. The normality of data was checked using boxplot, histogram, Q-Q plot, *Anderson-Darling test*, and *Shapiro-Wilk test*. To describe availability, following ranges were used: < 30% (very low), 30–49% (low), 50–80% (fairly high), and >80% (high) [22].

**Medicine prices.** To compare medicine prices among regions and countries, Median Price Ratio (MPR) for each medicine was calculated. The international reference prices (IRPs) were the median supplier prices taken from the 2015 International Medical Products Price Guide of Management Sciences for Health (MSH) [23]. For patient prices, MPRs should be lower than 1.5 in the public sector, and lower than 2.5 in the private sector [22].

$$Median\ Price\ Ratio = \frac{Median\ local\ unit\ price}{International\ reference\ unit\ price}$$

**Affordability.** The affordability of treatments for common diseases was computed as the number of days' wages of the lowest-paid unskilled government worker needed to purchase medicines prescribed at a standard dose. The daily wage used in the analysis was 102,333.3333 Vietnam dong per day (4.43US$) [24]. The treatment duration for an acute illness was the total duration of a full course of therapy while that for a chronic disease was 30 days. As per the WHO/HAI, an medicine can be considered affordable if its treatment course only costs one day's wage or less [8].

## Results

### The availability of EMs on the day of data collection

Regarding individual medicines, the availability of the following medicine groups in the public sector was far higher than that in the private sector, including narcotic and psychotropic medicines (morphine, diazepam), medicines used to treat diabetes mellitus (metformin, insulin), and cardiovascular diseases like hypertension and hypercholesterolemia (bisoprolol, captopril, simvastatin, atorvastatin, enalapril, furosemide). The availability of several medicines in the private sector was higher than that in the public sector including albendazole, mebendazole, salbutamol, paracetamol suspension, ibuprofen, and diclofenac. These medications were mainly used to treat common diseases (including parasitic worm infestation, pain, inflammation, and asthma). Moreover, in the private sector, the availability of OBs was lower than that of LPGs, excluding metformin, gliclazide, and salbutamol. Medicines whose availability was low in both sectors included amitriptyline, ceftriaxone, and paracetamol suspension (Table 1).

Although the average availability of OBs in the private sector was significantly higher than that in the public sector (p < 0.001, *Wilcoxon rank-sum test*), these values were extremely low (0.7% and 13.7%, respectively). By contrast, the mean availability of LPGs in the former was lower than that in the latter (p < 0.001, *Wilcoxon rank-sum test*). In both sectors, generic medicines were the predominant product type available. When the analysis was limited to survey medicines listed on the NEML, the mean availability of both OBs and LPGs was inconsiderably changed in both sectors (Table 2).

For regional analysis, the mean availability of OBs in Phuly city was the highest in both sectors. The mean availability of OBs in private medicine outlets in Phuly was significantly higher than that in Binhluc (p = 0.0065, *Dunn test*). In the private sector, the mean availability of LPGs in Phuly city was also the highest (Table 3). The mean availability of LPGs in public health facilities in Lynhan was significantly lower than that in Kimbang (p < 0.0001, *Dunn test*) and Duytien (p = 0.0206, *Dunn test*). In addition, the mean availability of both OBs and LPGs in urban areas was significantly higher than that in rural areas (p = 0.0013 and 0.0306, respectively, *Wilcoxon rank-sum test*) (Table 4).

**Table 1. The availability and median price ratio of each essential medicine.**

| No | Medicines | Availability (%) | | | | Median Price Ratio (MPR) | | | |
|---|---|---|---|---|---|---|---|---|---|
| | | Public sector | | Private sector | | Public sector | | Private sector | |
| | | OB | LPG | OB | LPG | OB | LPG | OB | LPG |
| | *Core medicines* | | | | | | | | |
| 1 | Amitriptyline 25 mg | - | 3.3 | - | 20.0 | - | - | - | 1.55 |
| 2 | Amoxicillin 500 mg | - | 83.3 | - | 74.3 | - | 0.67 | - | 1.15 |
| 3 | Bisoprolol 5 mg | 3.3 | 100 | 11.4 | 14.3 | - | 0.31 | 2.21 | 0.76 |
| 4 | Captopril 25 mg | - | 80 | - | 25.7 | - | 0.89 | - | 1.06 |
| 5 | Ceftriaxone 1 g | - | - | - | 22.9 | - | - | - | 2.62 |
| 6 | Ciprofloxacin 500 mg | - | 73.3 | 2.9 | 82.9 | - | 3.88 | - | 1.16 |
| 7 | Co-trimoxazole 48 mg/ml | - | 60.0 | - | 11.4 | - | 1.68 | - | 7.21 |
| 8 | Diazepam 5 mg | - | 100 | - | 2.9 | - | 2.74 | - | - |
| 9 | Diclofenac 50 mg | - | - | 25.7 | 74.3 | - | - | 38.47 | 2.40 |
| 10 | Metformin 500 mg | - | 90 | 34.3 | 2.9 | - | 1.75 | 5.34 | - |
| 11 | Omeprazole 20 mg | - | 83.3 | - | 100 | - | 0.39 | - | 1.23 |
| 12 | Paracetamol 24 mg/ml | - | - | - | 17.1 | - | - | - | 5.04 |
| 13 | Salbutamol 100 mcg/dose | - | 53.3 | 77.1 | 17.1 | - | 1.25 | 2.26 | 1.76 |
| 14 | Simvastatin 20 mg | - | 50 | - | 22.9 | - | 0.68 | - | 1.85 |
| | *Supplementary medicines* | | | | | | | | |
| 15 | Albendazole 200 mg | - | - | 28.6 | 31.4 | - | - | 14.29 | 10.21 |
| 16 | Amlodipine 5 mg | 3.3 | 100 | 2.9 | 100 | - | 0.90 | - | 1.37 |
| 17 | Atorvastatin 20 mg | - | 100 | 14.3 | 34.3 | - | 1.14 | 6.24 | 1.11 |
| 18 | Cefalexin 500 mg | - | 100 | - | 100 | - | 0.37 | - | 0.55 |
| 19 | Co-trimoxazole 480 mg | - | 96.7 | - | 97.1 | - | 0.76 | - | 1.08 |
| 20 | Enalapril 5 mg | - | 100 | - | 80.0 | - | 0.99 | - | 2.29 |
| 21 | Furosemide 40 mg | - | 83.3 | - | 82.9 | - | 0.88 | - | 3.55 |
| 22 | Gliclazide 30 mg | - | 50.0 | 51.4 | 11.4 | - | 1.29 | 4.44 | 1.33 |
| 23 | Ibuprofen 400 mg | - | - | - | 31.4 | - | - | - | 3.36 |
| 24 | Insulin 100 IU/ml | 12.5 | 100 | - | 8.6 | - | 0.72 | - | - |
| 25 | Mebendazole 500 mg | - | - | - | 88.6 | - | - | - | 8.51 |
| 26 | Metronidazole 250 mg | - | 100 | 57.1 | 77.1 | - | 1.49 | 10.64 | 2.31 |
| 27 | Morphine 10 mg/ml | - | 96.7 | - | - | - | 0.30 | - | - |
| 28 | Nifedipine retard 20 mg | - | 93.3 | 5.7 | 57.1 | - | 1.95 | - | 1.17 |
| 29 | ORS powder (1 liter) | -* | - | -* | 57.1 | - | - | - | 0.92 |
| 30 | Paracetamol 500 mg | - | 100 | 85.7 | 91.4 | - | 0.68 | 10.82 | 1.97 |

OB: originator brand, LPG: lowest-priced generic, IU: international unit, MPR: median price ratio.

*: The originator brand of ORS (Oral rehydration salts) cannot be found.

In four columns involving availability, "-"means "not available".

In four columns involving MPRs, "-"means "Medicines were not found in 4 facilities or more".

## Medicine prices

In the public sector, the prices of LPGs were nearly equal to their IRPs with median MPRs = 0.95 (25th–75th percentile: 0.69–1.44). By virtue of low availability, the prices of OBs were not found. LPGs priced several times higher than IRPs include ciprofloxacin (MPR = 3.88), diazepam (MPR = 2.74), and nifedipine (MPR = 1.95). In the private sector, OBs were generally sold at 6.24 times their IRPs (25th–75th percentile: 4.44–10.82) while this figure for LPGs was 1.65 (25th–75th percentile: 1.16–2.57). For OBs, MPRs of the following

**Table 2. The mean availability of medicines on the day of data collection.**

| Mean availability (standard deviation) | Public sector (n = 30 facilities) | | Private sector (n = 35 outlets) | |
|---|---|---|---|---|
| | OB | LPG | OB | LPG |
| of all 30 medicines | 0.7% (2.4%) | 63.2% (41.4%) | 13.7% (24.5%) | 47.9% (35.6%) |
| of 28 NEML medicines | 0.6% (2.5%) | 64.2% (40.5%) | 14.3% (25.3%) | 50.2% (35.8%) |
| of 14 global medicines | 0.2% (0.9%) | 55.5% (38.9%) | 10.8% (22.0%) | 34.9% (32.7%) |
| of 16 supplementary medicines | 1.1% (3.3%) | 70.0% (43.5%) | 16.4% (27.1%) | 59.3% (35.1%) |

OB: originator brand, LPG: lowest-priced generic, NEML: national essential medicines list.

**Table 3. Regional analysis: Comparison of the mean availability of EMs across the six regions surveyed.**

| Type | Mean availability | | | | | | p-value* |
|---|---|---|---|---|---|---|---|
| | Phuly | Duytien | Thanhliem | Binhluc | Lynhan | Kimbang | |
| **Public sectors** | | | | | | | |
| OBs | 2.5% | 0% | 0% | 0% | 0% | 0% | 0.4159 |
| LPGs | 62.7% | 65.5% | 62.1% | 65.5% | 58.6% | 72.4% | 0.00028 |
| **Private sectors** | | | | | | | |
| OBs | 24.8% | 17.8% | 12.6% | 7.9% | 10.3% | 10.9% | 0.00637 |
| LPGs | 64.7% | 46.7% | 46.7% | 42.4% | 47.3% | 43.3% | 0.1778 |

OB: originator brand, LPG: lowest-priced generic.

* p-value was calculated using Kruskal-Wallis rank-sum test.

**Table 4. Private sector: Comparison of the mean availability of EMs between urban and rural areas.**

| Medicine group | Mean availability (SD) | | p-value** |
|---|---|---|---|
| | Urban area (n = 11 medicine outlets) | Rural area (n = 24 medicine outlets) | |
| OBs | 20.06% (7.37%) | 10.78% (6.38%) | p = 0.001314 |
| LPGs | 57.27% (12.37%) | 43.61% (9.42%) | p = 0.03063 |
| All medicines* | 66.97% (11.87%) | 48.75% (10.85%) | p = 0.0002384 |

OB: originator brand, LPG: lowest-priced generic.

*: A medicine was available in a medicine outlet when its OB or/and LPG was found.

**: p-value was calculated using Wilcoxon rank-sum test.

medicines were extremely high: diclofenac (38.47), albendazole (14.29), paracetamol (10.82), and metronidazole (10.64). LPGs priced several times higher than IRPs include albendazole (MPR = 10.21), mebendazole (MPR = 8.51), cotrimoxazole suspension (MPR = 7.21), and paracetamol suspension (MPR = 5.04). Only 5 LPGs in the public health facilities had MPRs of more than 1.5, and 7 LPGs in private medicine outlets had MPRs of more than 2.5. Furthermore, the 25th and 75th percentiles for individual medicines showed that: for LPGs, in both sectors, prices did not significantly vary among health facilities/private medicine outlets but for OBs, in the private sector, prices significantly varied among medicine outlets (Tables 1 and 5).

In private medicine outlets, for eight medicines found as both product types (OBs and LPGs), OBs cost 351.34% more than their generic equivalents (S3 Table). Comparison of the prices of OBs and LPGs cannot be done in the public sector because of the unavailability of almost OBs. For 18 LPGs found in both sectors, the final patient prices in the private sector were 35.4% higher than those in the public sector (S4 Table). For regional analysis, in the

private sector, median MPRs for OBs ranged from 6.09 in Thanhliem district to 10.64 in Lynhan district. The median MPRs for generics in both sectors did not differ significantly across six surveyed areas: from 0.72 to 1.36 in the public sector and from 1.18 to 1.74 in the private sector (Table 5).

## Affordability

The affordability of LPGs in both sectors was good for all conditions, with standard treatments costing only a day's wage or less. Regarding OBs, treatments costing over a days' wage of the lowest-paid government worker in the private sector include hypertension (bisoprolol, 2.7 days), arthritis (diclofenac, 2.3 days), and diabetes (metformin, 1.6 days) (Table 6).

**Table 5. Regional analysis: Median MPRs of EMs across the six regions surveyed.**

| Type | Hanam province | Phuly city | Duytien district | Thanhliem district | Binhluc district | Lynhan district | Kimbang district |
|---|---|---|---|---|---|---|---|
| **Public sector** | | | | | | | |
| OBs | - | - | - | - | - | - | - |
| LPGs | 0.95 | 0.74 | 1.00 | 1.36 | 0.72 | 1.29 | 0.88 |
| **Private sector** | | | | | | | |
| OBs | 6.24 | 6.73 | 7.14 | 6.09 | 7.08 | 10.64 | 6.54 |
| LPGs | 1.65 | 1.18 | 1.74 | 1.74 | 1.73 | 1.30 | 1.38 |

OB: originator brand, LPG: lowest-priced generic.

"-"means "There were lower than four MPRs of OBs found".

**Table 6. Number of days' wages of the lowest-paid government worker needed to purchase standard treatments.**

| Disease condition and standard treatment | | | Day's wages to pay for treatment | | |
|---|---|---|---|---|---|
| Condition | Medicine name, strength, dosage form | Treatment schedule | LPG—public sector | LPG–private sector | OB–private sector |
| Asthma | Salbutamol 100 mcg/dose inhaler | 1 inhaler of 200 doses | 0.5 | 0.7 | 0.9 |
| Diabetes | Metformin 500 mg cap/tab | 1 cap/tab x 3 x 30 days = 90 cap/tab | 0.5 | - | 1.6 |
| Hypertension | Bisoprolol 5 mg cap/tab | 1 cap/tab x 2 x 30 days = 60 cap/tab | 0.4 | 0.9 | 2.7 |
| Hypertension | Captopril 25 mg cap/tab | 1 cap/tab x 2 x 30 days = 60 cap/tab | 0.3 | 0.4 | - |
| Hypercholes-terolaemia | Simvastatin 20 mg cap/tab | 1 cap/tab x 30 days = 30 cap/tab | 0.2 | 0.7 | - |
| Depression | Amitriptyline 25 mg cap/tab | 1 cap/tab x 3 x 30 days = 90 cap/tab | - | 0.3 | - |
| Paediatric respiratory infection | Co-trimoxazole 8+40 mg/ml suspension | 5 ml twice a day x 7 days = 70 ml | 0.1 | 0.5 | - |
| Adult respiratory infection | Ciprofloxacin 500 mg cap/tab | 1 cap/tab x 2 x 7 days = 14 cap/tab | 0.5 | 0.1 | - |
| Adult respiratory infection | Amoxicillin 500 mg cap/tab | 1 cap/tab x 3 x 7 days = 21 cap/tab | 0.1 | 0.2 | - |
| Adult respiratory infection | Ceftriaxone 1 g/vial injection | 1 vial | - | 0.2 | - |
| Anxiety | Diazepam 5 mg cap/tab | 1 cap/tab x 7 days = 7 cap/tab | 0.04 | - | - |
| Arthritis | Diclofenac 50 mg cap/tab | 1 cap/tab x 2 x 30 days = 60 cap/tab | - | 0.1 | 2.3 |
| Pain/inflammation | Paracetamol 24 mg/ml suspension | Child one year: 120 mg (= 5ml) x 3 for 3 days = 45 ml | - | 0.3 | - |
| Ulcer | Omeprazole 20 mg cap/tab | 1 cap/tab x 30 days = 30 cap/tab | 0.04 | 0.1 | - |

OB: originator brand, LPG: lowest-priced generic, cap/tab: capsule/tablet, mcg: microgram.

"-"means "Medicines were not found in 4 facilities or more".

All above treatment schedules were taken from documents of the WHO/HAI [8, 18].

### Comprehensive analysis of the availability and prices of LPGs

Figs 1 and 2 display the availability and MPRs of LPGs in the public and private sectors, respectively. There are four quadrants in each figure. Quadrant IV (bottom-right quadrant) contains EMs with high availability and low MPRs. In the public sector, most of the EMs are in Quadrant IV while in the private sector, this quadrant only contains six EMs. In addition, Quadrant II (top-left quadrant) contains EMs with low availability and high MPRs. People can have difficulties in accessing and affording these medicines. There are two and five EMs in Quadrant II, respectively for the public sector and the private sector. It is remarkable that in both sectors, all surveyed suspensions (one type of child-friendly formulations) are in Quadrant II (including co-trimoxazole 40+8 mg/ml and paracetamol 24 mg/ml). Paracetamol suspension 24 mg/ml is absent in Fig 1 by reason of its unavailability in the public sector. Their alternative formulations (co-trimoxazole tablet 400+80 mg and paracetamol tablet 500 mg) are in Quadrant IV in both sectors.

## Discussion

### The availability of EMs

In Hanam, the mean availability of OBs was low in both sectors (public sector 0.7%, private sector 13.7%). Due to the variation in the list of surveyed medicines, there are difficulties in equal comparing our results to previous studies. However, in Hanam, the mean availability of OBs in both public and private sector was respectively lower than that in Jordan (2016): 9%

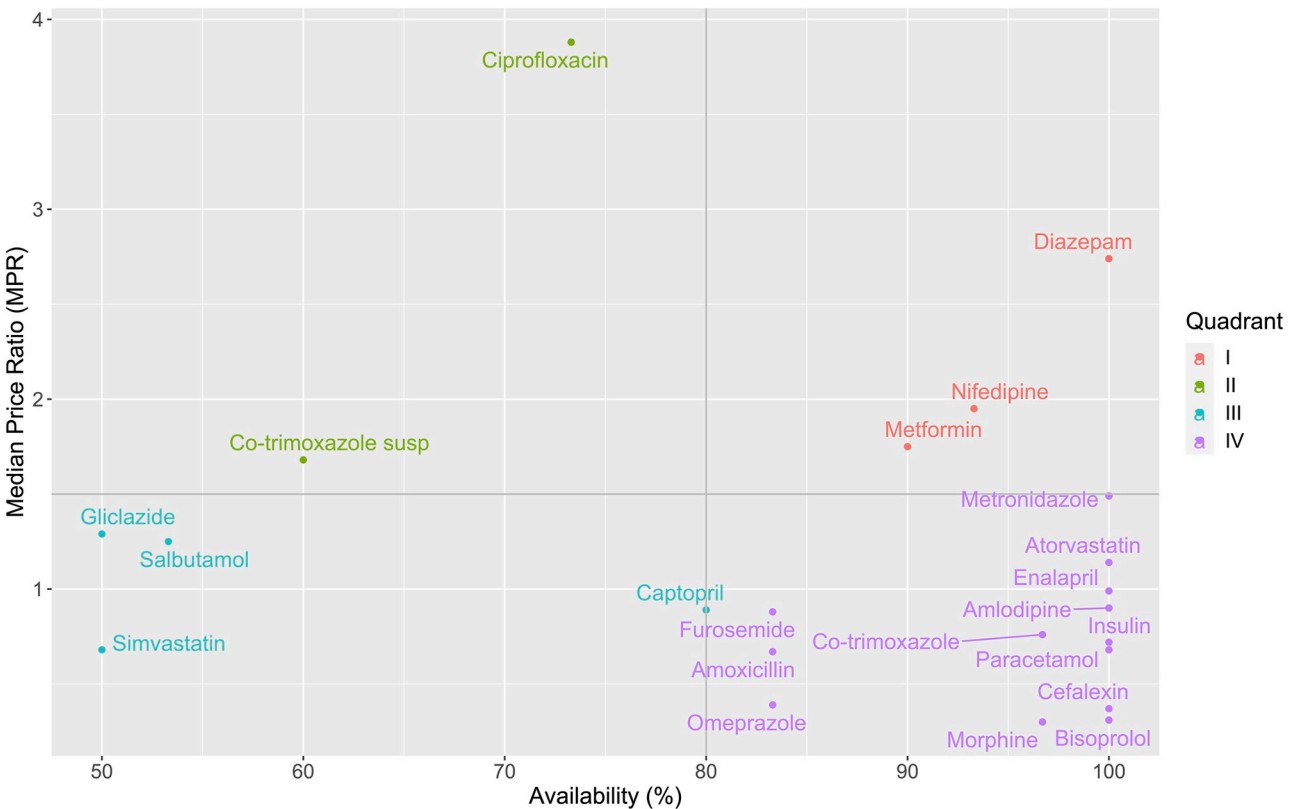

**Fig 1. Comprehensive analysis of the availability and prices of LPGs in the public sector.** Horizontal line: MPR = 1.5, Vertical line: Availability = 80%, susp: suspension.

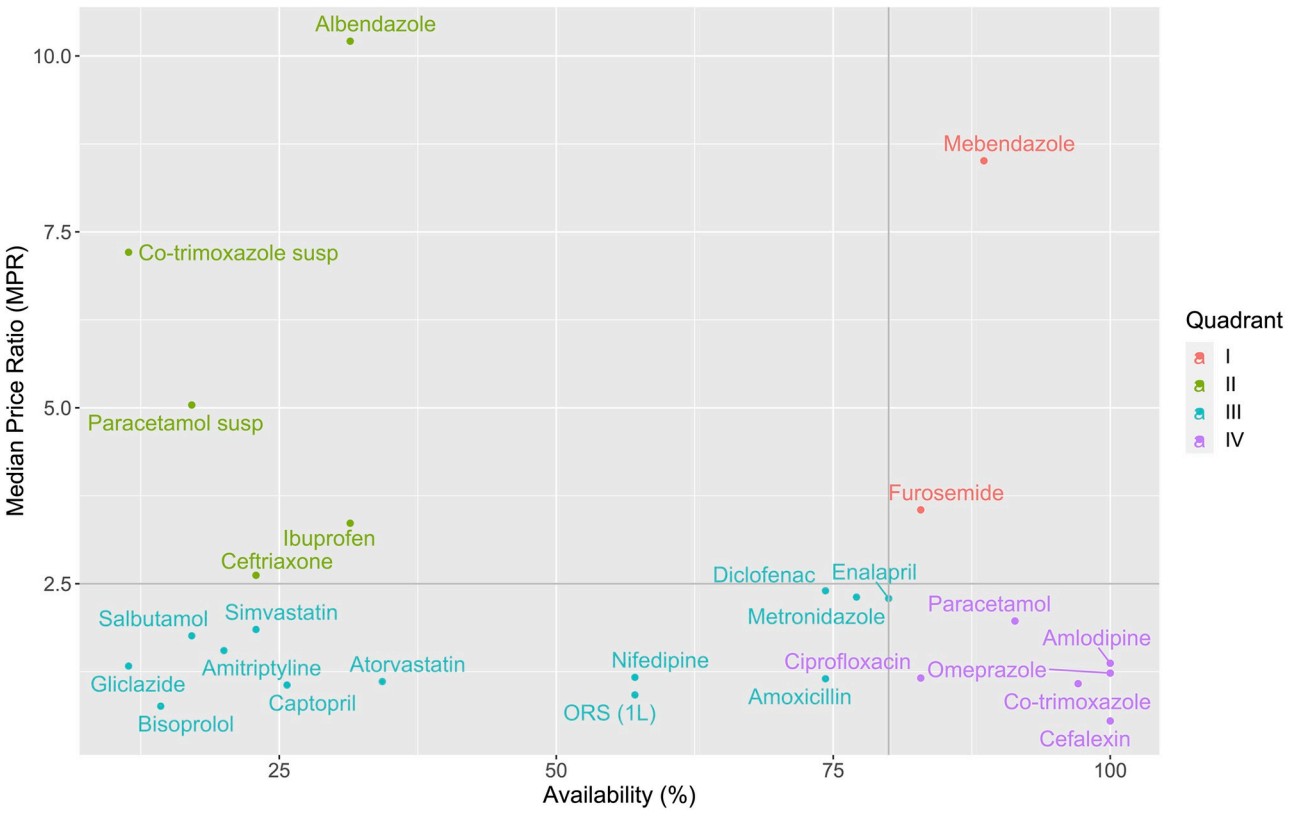

**Fig 2. Comprehensive analysis of the availability and prices of LPGs in the private sector.** Horizontal line: MPR = 2.5, Vertical line: Availability = 80%, susp: suspension.

and 57% [25]; Lahore Division, Pakistan (2016–2017): 6.8% and 55.0% [26]; Zhejiang, China (2018): 41.8% and 35.1% [27]; Rwanda (2019): 11.7% and 29.2% [28]; higher than results from a study conducted in Bangladesh (2015): 0% and 4% [29]. In addition, the mean availability of LPGs in Hanam was fairly high in the public sector (63.2%) but still low in the private sector (47.9%). These results were lower than findings from Iran (2014): 75.5% and 83.3% [30], Malaysia (2017): 74.8% and 49.1% [31]; higher than results of Lahore Division, Pakistan (2016–2017): 35.3% and 20.3% [26], and Zhejiang, China (2018): 35.1% and 40.3% [27], respectively for the public sector and the private sector.

In general, the Hanam healthcare system can adequately supply EMs to meet inhabitants' needs. By virtue of low costs and national policies, generic medicines were usually prioritized in use in the public sector. Accordingly, the availability of OBs was low and patients were mainly treated with LPGs. In the private sector, the availability of morphine and diazepam was far lower than that in the public sector. These medicines are restricted in use and only a few private medicine outlets with special permission are eligible to sell them [32]. Regarding medicines used to treat diabetes (metformin and insulin) and cardiovascular diseases including hypertension (bisoprolol, captopril, enalapril, furosemide) and hypercholesterolemia (simvastatin, atorvastatin), their availability in the public sector was far higher than that in the private sector since these medicines were covered by health insurance. In public health facilities, patients with a health insurance card can nearly get these medicines free of charge. For albendazole and mebendazole, in Vietnam, the government annually conducts deworming campaigns for children. This can be a reason explaining the low availability of these anthelmintics in public facilities.

In the private sector, the high prices of OBs, the low demands, and the low incomes of inhabitants can be several reasons explaining why the mean availability of OBs was low. In addition, thanks to the development of transportation and logistics systems, pharmacists said that although numerous medicines were unavailable in their medicine outlets, patients/customers could order various kinds of medicines they need and get them in the next several days. This can be the important rationale behind the low availability of many medicines in private medicine outlets, especially those covered by health insurance (for example simvastatin and bisoprolol). Although the costs of Ventolin (salbutamol), Diamicron (gliclazide), and Glucophage (metformin) were quite high, the availability of these OBs was far higher than that of LPGs. Three aforementioned OBs were popularly used in Vietnam because of their outstanding quality and efficiency.

Among six surveyed areas, the mean availability of EMs in the private sector of Phuly was the highest because this city is the center of Hanam. The mean availability of EMs in the rural areas was significantly lower than that in the urban areas. This can be a matter of concern because the countryside constitutes a majority of areas in many provinces of Vietnam. In addition, in rural areas, many people (especially senior citizens) are living below the poverty line. The low availability and high prices can be barriers hindering their access to EMs. As per WHO, the availability of EMs should be at least 80% [33, 34]. However, in this study, the availability of 11 EMs was lower than this benchmark. The government should have useful plans to expand access to EMs in the Vietnamese population, especially impoverished people living in the countryside.

## Medicine prices and affordability

In Hanam, the prices of LPGs in the public sector were acceptable with median MPRs = 0.95, compatible with findings in Jordan (1.16) [25], Rwanda (1.0) [28], Shaanxi, China (1.49) [35] but far lower than the result from Zhejiang, China (5.21) [27]. In the private medicine outlets of Hanam, OBs and LPGs were sold at 6.24 and 1.65 times their IRPs which were lower than the findings in Jordan (2016): 9.7 and 7.5 [25] and Zhejiang, China (2018): 14.75 and 4.94 [27], respectively for OBs and LPGs. OBs cost 351.34% more than LPGs in Hanam, lower than the result from Kenya (2016): innovator brands were 13.8 times more expensive than generic medicines [36]. In Hanam, among surveyed areas, the prices of LPGs did not differ significantly. By contrast, findings from a study conducted in five provinces of China (2010–2018) show that the variation was significant for medicine prices across provinces [37].

There were several child-friendly medicines investigated in Hanam. For paracetamol suspension 24mg/ml, the availability of this formulation was extremely low in both sectors. High cost is one reason (MPR = 5.04). In northern Ethiopia (2016), this medicine was even sold at 19.4 times and 26.2 times higher than its IPR, respectively for the public and private sectors [38]. There are several child-friendly formulations of paracetamol ubiquitously used in Vietnam (for example powder packet 150 mg and 250 mg). Unfortunately, their IRPs cannot be found in the 2015 International Medical Products Price Guide of MSH [23]. As a consequence, they cannot be surveyed in this study. Similar to paracetamol, the availability of co-trimoxazole suspension 40+8 mg/ml and albendazole 200 mg was low while their MPRs were high. The low availability and high prices of pediatric medicines were also reported in studies from Mongolia (2016) [39], Jiangsu, China (2017) [40], and Ethiopia (2018–2019) [41]. These three medicines are all in Quadrant II (low availability and high MPRs) in the comprehensive analyses of EMs' availability and prices. Children in Hanam may have difficulties in access to formulations that are suitable for them (such as suspensions and chewable tablets).

In Hanam, OBs were unaffordable but the affordability of LPGs in both sectors was good for all conditions: standard treatments costing only a day's wage or less. Our findings were in line with results from Iran (2014) [30], Nepal (2015) [42], Anhui, China (2015) [43], Pakistan (2016–2017) [26], Malaysia (2017) [31], and Zhejiang, China (2018) [27]. The results from a study in 11 countries of the Asia Pacific Region also showed that buying a month's supply of LPGs required less than one day's wage in most countries [44]. However, in Jordan, Zambia, and Ethiopia, medicine prices were not affordable [25, 45, 46]. In Malawi (2017), Cameroon, and Congo (2017–2018), the cost for one standard treatment of more than a half surveyed medicines exceeded the daily wage, making them unaffordable to a multitude of the population [47, 48].

In the light of low availability, OBs were not a major feature of the pharmaceutical market in Hanam. As a result, it seems that unaffordability issues regarding OBs were insubstantial. Generally, the availability of generic medicines was fairly high and their prices were affordable. In recent years, the Vietnamese government has implemented many national policies to increase the availability of generic medicines and reduce medicine prices [49, 50]. Wholesale and retail prices of medicines were posted up at transaction or medicine-selling places. The maximum retail surplus for medicines sold at medicine retails within medical examination and treatment establishments was set. The declared wholesale and retail prices of medicines (at medicine trading establishments) and winning-bid prices of medicines used in public health facilities were published on the Ministry's e-portal [49]. In public facilities, a majority of medicines covered by health insurance are EMs. Furthermore, in 2019, the Vietnam Ministry of Health promulgated a list of 640 medicines that domestic pharmaceutical companies would have the capacity for manufacturing and supplying. This list mainly involves medicine-tendering activities in public health facilities. For these 640 medicines, imported medicines having the same active ingredients, dosage forms, and doses are restricted in the bidding process and public health facilities mainly use low-priced medicines locally manufactured. In private medicine outlets, regarding medicine prices, the government only requires that all medicines stocking in any medicine outlet must be labeled with prices on the medicine containers (boxes). In order to monitor and control the quality of medicines, the Department of Health in each province periodically inspects and surveys private medicine outlets. Many samples of medicines are randomly selected and tested to assess their quality.

## Limitations

Regarding EMs, this is the first comprehensive study using the WHO/HAI methodology to measure the availability, prices, and affordability of their OBs and LPGs in Hanam, Vietnam. By reason of the COVID-19 outbreak, the travel restrictions, and the paucity of funding and human resources, we are only able to conduct this research in one province devoid of COVID-19 patients, not for the whole country. This study had some limitations. Firstly, data on EMs were collected on the day of data collection, not reflect the average availability over time. Only 30 surveyed EMs cannot reflect the whole EMs on the market. As per WHO's instructions, medicines' availability was reported as mean and standard deviation although data was not normally distributed. Some medicines (such as paracetamol, gliclazide, metformin) were found in different strengths, so the low availability of these medications may not be meaningful. In addition, only medicines with an MSH IRP could be included in this survey. The data was collected in 2020 but the latest IRPs are for the year 2015. Last but not least, affordability was computed based on the government's lowest daily wage. However, many inhabitants could earn less than that value, especially farmers in rural areas.

## Conclusions

The availability of OBs was significantly low in both the public and private sectors. In private medicine outlets, the prices of OBs were high and they were unaffordable. In both sectors, generic medicines were the predominant product type available. The mean availability of LPGs was fairly high but still lower than the benchmark of WHO. LPGs in both sectors were sold to patients at reasonable prices compared to IRPs. A national-scale study should be conducted in the forthcoming years to provide a comprehensive picture of the availability, prices, and affordability of EMs, thereby helping the Vietnamese government to identify the urgent priorities and improving access to EMs.

## Supporting information

**S1 Data.**
(XLS)

**S1 Table. Surveyed areas and the number of health facilities from which data were collected in this research.**
(DOCX)

**S2 Table. The list of surveyed medicines.**
(DOCX)

**S3 Table. Median MPRs for eight medicines found as both product types in the private sector.**
(DOCX)

**S4 Table. Median MPRs for medicines found in both public and private sectors.**
(DOCX)

## Acknowledgments

The authors would like to thank Mr. Kha Quach Xuan, a leader of the Hanam Department of Health assisting us in the process of data collection. We also want to express our gratitude to all pharmacists working in health facilities selected for investigation in this research.

## Author Contributions

**Conceptualization:** Huong Thi Thanh Nguyen, Dai Xuan Dinh, Trung Duc Nguyen, Van Minh Nguyen.

**Data curation:** Dai Xuan Dinh.

**Formal analysis:** Dai Xuan Dinh.

**Investigation:** Dai Xuan Dinh, Van Minh Nguyen.

**Methodology:** Huong Thi Thanh Nguyen, Dai Xuan Dinh, Trung Duc Nguyen, Van Minh Nguyen.

**Project administration:** Huong Thi Thanh Nguyen, Dai Xuan Dinh.

**Software:** Dai Xuan Dinh.

**Supervision:** Huong Thi Thanh Nguyen, Trung Duc Nguyen.

**Validation:** Huong Thi Thanh Nguyen.

**Visualization:** Dai Xuan Dinh.

**Writing – original draft:** Dai Xuan Dinh.

**Writing – review & editing:** Huong Thi Thanh Nguyen, Dai Xuan Dinh, Trung Duc Nguyen, Van Minh Nguyen.

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
