## [Decision Letter · Decision Letter 0]

26 Jul 2021

PONE-D-21-15309

Availability, prices and affordability of essential medicines: a cross-sectional survey in Hanam province, Vietnam

PLOS ONE

Dear Dr. Dinh,

Thank you for submitting your manuscript to PLOS ONE. After careful consideration, we feel that it has merit but does not fully meet PLOS ONE’s publication criteria as it currently stands. Therefore, we invite you to submit a revised version of the manuscript that addresses the points raised during the review process.

We look forward to receiving your revised manuscript.

Kind regards,

Caroline Anita Lynch

Academic Editor

PLOS ONE

2. Please include in your Methods section (or in Supplementary Information files) the participating hospitals/institutions.

3. Please provide additional details regarding participant consent. In the ethics statement in the Methods and online submission information, please ensure that you have specify:

a) whether the ethics committee approved the verbal/oral consent procedure,

b) why written consent could not be obtained, and 3) how verbal/oral consent was recorded. If your study included minors, please state whether you obtained consent from parents or guardians in these cases. If the need for consent was waived by the ethics committee, please include this information.

Additional Editor Comments (if provided):

Please could you insert a line in the introduction to describe the utilization of public versus private sector in Vietnam as well as addressing reviewers comments.

Reviewers' comments:

Reviewer's Responses to Questions

**Comments to the Author**

1. Is the manuscript technically sound, and do the data support the conclusions?

Reviewer #1: Partly

Reviewer #2: Yes

2. Has the statistical analysis been performed appropriately and rigorously? 

Reviewer #1: Yes

Reviewer #2: Yes

3. Have the authors made all data underlying the findings in their manuscript fully available?

Reviewer #1: Yes

Reviewer #2: Yes

4. Is the manuscript presented in an intelligible fashion and written in standard English?

Reviewer #1: Yes

Reviewer #2: Yes

5. Review Comments to the Author

Reviewer #1: Evaluation

Thanks for sharing this manuscript (Availability, prices and affordability of essential medicines: a cross-sectional survey in Hanam province, Vietnam), which contains interesting information. The authors raised very important issue (access to essential medicine) in the this work; however I have minor issues in three areas in the manuscript that need to be addressed before it is accepted for publication:

1. Introduction

In lines 66-67, you stated several studies on the matter has been conducted; so what makes your different and urgent (is that because previous woks were unpublished?) what does it mean “small-scale studies”? (Lines 67 and 68)

The introduction part does not introduce pharmaceutical delivery system of the Vietnam (Good to know here the current situation regarding healthcare in Vietnam, e.g. how is healthcare typically provided. What are the current co-payments for visiting a physician, e.g. 100%, 50%, etc., and what is this amount relative to the daily wage of the LPGW? What about medicines - are these 100% co-pay - some medicines are free for the public in the public sector and covered through insurance - what are these medicines and how are they chosen? ….do public hospitals and primary healthcare centers stock and dispense similar items? Are all drug outlets in private sectors drug stores and possess all types of EMs that public hospitals and primary healthcare centers can—if there is level of capacity of handling medicine?...), challenges to achieve this paper’s interest, reasons that trigger this work, and previous studies findings.

2. The method

Why was the Hanam chose over other potential sources in Vietnam?

How many districts are there in Hanam, and how are these six districts selected?

Are all the medicines included in this study expected to be equally available in hospitals, primary healthcare centers and drug stores due to legal restriction? If not, this may give wrong information about the availability of the medicines.

Since you are informing your study objective, how do you obtain/trace/ real patient price data on medicines especially from private sectors?

For patient prices, MPRs should be lower than 1.5 in the public sector, and lower than 2.5 in the private sector. (Lines 129-130) Is there an agreed cut-point?

The daily wages of the lowest-paid unskilled government worker in USD during data collection (lines 135-136)

Affordability limit/declaration cut-off this work?

3. Results

Table 1 has to list individual medicines included in this study. The authors search continent approach to incorporate first column [Mean availability (standard deviation - SD)] in their new table. Such modification will give more detailed information about availability than the general one.

Table 4. For price comparison, as the guideline, it is recommended to take medicines found at least four of drug outlets in each sites from both sectors. In current work OBs are absent in the public sector, no need to use them. The remaining LPGs found in both sectors has to be listed individually and their MPRs (Min, 25th %, median, 75th % and Max) included. Focus your comparative discussion on these drugs only. If possible, try to merge, Table 5 in to Table 4. Figures that will be listed in the revised Table 1 may show the rest.

Table 5 would particularly benefit if it is changed into Figure for better general analysis. Using your reference #13 patient price cut-off point, you can point out patient prices of selected medicine. There are now standard graphics (available from HAI) in which both availability and affordability are combined in one graph. The x-axis shows availability (ranging from 0-100%, with a vertical line at 80%. The y-axis shows the affordability, with a horizontal line at the chosen cut-off point. In this graph all products have a point-position. In an ideal world all products are located in the right-lower corner of the graph, with availability at >80% and affordability below cut-off. The graph can clearly visualize where the problems are.

Table 6 would particularly benefit from an extensive review. In column 2, Drug name, strength, dosage form, dose, route of administration, frequency & treatment duration, Colum 3, Treatment schedule, The total amount of drug required to cover the complete treatment regimen, Column 4, Average drug Price per Unit (USD), and Column 5, Number of day’s wage to pay for treatment. In addition, sate reasons/description why bisoprolol 5 mg cap/tab and captopril 25 mg cap/tab used for hypertension management. This briefly shows how Vietnam STG (standard treatment guideline) looks like and how closer/far to/from IPRs (cost wise).

(See these references for Figure https://doi.org/10.1186/s12889-021-10745-5 and https://doi:10.1371/journal.pone.0070836)

Reviewer #2: Thank you for the opportunity to review this scholarly manuscript. The erudite disposition of the authors is commendable.

The paper has claimed that the availability of OBs was significantly low in both public and private sectors. Generic medicines were the predominant product type available in both public and private sectors. The mean availability of LPGs was fairly high but still lower than the benchmark of WHO. The prices of OBs were high and were unaffordable in private drug stores. LPGs in both sectors were sold to patients at reasonable prices compared to International retail prices.

These claims are properly placed in the context of available literature and the literature were fairly treated. Also, the data and analysis fully support the claim. A few minor remarks were noted. These would require the authors’ attention.

Abstract

- Result (line 33): insert the exact p value.

Introduction

Line 50: Insert reference.

Line 66: ………There are several studies on medicine prices, availability and affordability conducted in Vietnam [11, 12]. Some small scale studies were carried out but not published……….

What then is the gap in knowledge that this current study aimed to fill? Does this suggest that these studies were not on Essential medicines

Materials and methods

Lines 86/87: ………A licensed private drugstore closest to each of the selected public facilities was selected for the private sector………

The WHO/HAI methodology also provides for survey of 'Other' pharmacies which include those domiciled in private/organization owned hospitals. Does the province have these categories of hospitals? if yes, why were they not included in the survey? if no, a brief description of the health facility set up in Hanam would suffice.

Line 90: write ‘30’ in words

Results:

Table 3: Urban (11 drug stores) versus rural (24 drug stores) comparison

Does this suggest that most of the districts surveyed were in the rural area? A brief sentence regarding the geography of these areas should be added in the methods section

S1 Table

It is suggested that these medicines be group according to their class of drug. For example: Antibiotics, Psychotropic, Narcotics, Anti-inflammatory, Analgesics, Asthma medicines, Antihypertensive, Antidaibetic etc

Discussion

Line 215: …….fairly high (public sector 63.2%, private sector 47.9%)……….

Recast to reflect that availability of LPGs in private sector was fairly high but that in public sector was low (47.9%)

Line 283: Vietnamese government has implemented many national policies to increase the availability reduce medicine prices

The authors should stress the need for the enforcement of the price regulation in the private sector. Despite that fact that prices are generally lower than those of other countries, the private sector still sells OBs higher than the LPGs to the tune of 351.34%. Unless the LPGs are of reasonable quality and efficacy in treatment of the particular disease. Are there concerns for substandard medicines in Hanam?

6. PLOS authors have the option to publish the peer review history of their article (what does this mean?). If published, this will include your full peer review and any attached files.

Reviewer #1: **Yes: **Habtamu Abuye Lambore

Reviewer #2: No

---

## [Author Response · Author response to Decision Letter 0]

1 Sep 2021

We strived to follow these templates.

2. Please include in your Methods section (or in Supplementary Information files) the participating hospitals/institutions.

Information on surveyed main hospitals and health facilities was added in the S2 Table and Method section (lines 136 - 147).

3. Please provide additional details regarding participant consent. In the ethics statement in the Methods and online submission information, please ensure that you have specify:

a) whether the ethics committee approved the verbal/oral consent procedure,

b) why written consent could not be obtained, and 

c) how verbal/oral consent was recorded. If your study included minors, please state whether you obtained consent from parents or guardians in these cases. If the need for consent was waived by the ethics committee, please include this information.

The verbal consent procedure was approved by the ethics committee of Hanoi University of Pharmacy. The ethics committee did not require the research team to gain written consent from interviewees because in this study, we did not do clinical trials, did not take blood samples, and did not do any activities which could harm interviewees in any form. We only asked pharmacists about the availability and prices of medicines and therefore, it seems that the data-collection process did not have any detrimental effects on interviewees. 

Not that written consent could not be obtained. From the beginning, we did not intend to obtain written consent. All public health facilities and private medicine outlets in our research are managed by the Hanam Department of Health. As a result, when data collectors gave a letter of endorsement (from the Hanam Department of Health) to interviewees (pharmacists), they were willing to participate in our research and answered questions about the availability and prices of essential medicines with pleasure. If there is anyone who did not want to take part in our research, we could not collect the data of their facilities. 

Reviewer 1

1. Introduction

In lines 66-67, you stated several studies on the matter has been conducted; so what makes your different and urgent (is that because previous woks were unpublished?) what does it mean “small-scale studies”? (Lines 67 and 68)

We adjusted these sentences. The new information can be seen in lines 89 - 95. In the process of seeking previous studies involving essential medicines in Vietnam, we found several studies (written in the Vietnamese language). “Small-scale studies” in the first manuscript means in these studies, researchers only surveyed private medicine outlets (a study conducted in 2016), the low sample size - 14 public health facilities(a study conducted in 2014)... All these studies were conducted before the year 2017.

The introduction part does not introduce pharmaceutical delivery system of the Vietnam (Good to know here the current situation regarding healthcare in Vietnam, e.g. how is healthcare typically provided. What are the current co-payments for visiting a physician, e.g. 100%, 50%, etc., and what is this amount relative to the daily wage of the LPGW? What about medicines - are these 100% co-pay - some medicines are free for the public in the public sector and covered through insurance - what are these medicines and how are they chosen? ….do public hospitals and primary healthcare centers stock and dispense similar items? Are all drug outlets in private sectors drug stores and possess all types of EMs that public hospitals and primary healthcare centers can—if there is level of capacity of handling medicine?...), challenges to achieve this paper’s interest, reasons that trigger this work, and previous studies findings.

We did endeavor to supply information as much as possible.

The introduction to the Vietnam healthcare system can be seen in lines 64 - 91.

2. The method

Why was the Hanam chose over other potential sources in Vietnam?

How many districts are there in Hanam, and how are these six districts selected?

The introduction to the Hanam province and the reasons explaining why this province was chosen for investigation were added (lines 114 - 131). Hanam is a small province that is contiguous to the Hanoi capital. This province includes one city and five districts. All six areas were selected for investigation. All areas met the requirements of the WHO/HAI.

Are all the medicines included in this study expected to be equally available in hospitals, primary healthcare centers and drug stores due to legal restriction? If not, this may give wrong information about the availability of the medicines.

For the private sector, all medicines included in this study were expected to be equally available in private medicine outlets. All essential medicines are allowed to be stocked and sold in medicine outlets.

For the public sector, several medicines were only expected to be available in high-level hospitals/health centers (There are three levels: provincial, district, and commune). These medicines can be seen in the column “Levels of care” in the sheet “Reference prices” from the Workbook (S1 Data file). For these medicines, their availability was automatically computed and adjusted by the WHO/HAI workbook. For example, the level-of-care of insulin was 2. It means this medicine was only expected to be available in provincial and district health facilities (8 facilities). In our study, this medicine was available in only one public health facility (originator brand - OB). Therefore, the availability of OB insulin was equal to 1/8 (12.5%), in lieu of 1/30. 

Since you are informing your study objective, how do you obtain/trace/ real patient price data on medicines especially from private sectors?

Interviewees (pharmacists) were only informed that some researchers would go to health facilities to do a scientific study (informed by leaders of the Hanam Department of Health and leaders of district health centers). We only informed interviewees about our study objective when we paid a visit to surveyed facilities to collect data. 

The patient prices were taken from the labels of medicine containers (boxes) or computers. As per the Vietnam Pharmacy Law, all medicines in any medicine outlet must be labeled with a price tag on the boxes. The Department of Health periodically inspects medicine outlets. If there is any medicine without the price on the box, the owner will be fined. In the process of data collection, we did not observe any drugstores possessing a medicine without the price on the box. In some big medicine outlets and public health facilities, medicine prices were stored on the computers and we collected these prices from the computers.

For patient prices, MPRs should be lower than 1.5 in the public sector, and lower than 2.5 in the private sector. (Lines 129-130) Is there an agreed cut-point?

In many previous studies, these cut-off points were used to assess the suitability of medicine prices. For example:

- Gelders S, Ewen M, Noguchi N, Laing R. Price, availability and affordability: an international comparison of chronic disease medicines. World Health Organization and Health Action International; 2006. (Reference 22).

- Tadesse T, Abuye H, Tilahun G. Availability and affordability of children essential medicines in health facilities of southern nations, nationalities, and people region, Ethiopia: key determinants for access. BMC Public Health. 2021;21(1):714. doi: 10.1186/s12889-021-10745-5. PMID: 33849513; PMCID: PMC8045262. (Reference 46)

- Jiang M, Yang S, Yan K, Liu J, Zhao J, Fang Y (2013) Measuring Access to Medicines: A Survey of Prices, Availability and Affordability in Shaanxi Province of China. PLoS ONE 8(8): e70836. https://doi.org/10.1371/journal.pone.0070836

- ...

In our study, we used these cut-off points too.

The daily wages of the lowest-paid unskilled government worker in USD during data collection (lines 135-136)

Affordability limit/declaration cut-off this work?

The cut-off point was added (one day’s wage) in lines 192 - 194.

3. Results

Table 1 has to list individual medicines included in this study. The authors search continent approach to incorporate first column [Mean availability (standard deviation - SD)] in their new table. Such modification will give more detailed information about availability than the general one.

A table with the availability and MPRs of individual medicines was added (Table 1 in the revised manuscript).

Table 4. For price comparison, as the guideline, it is recommended to take medicines found at least four of drug outlets in each sites from both sectors. In current work OBs are absent in the public sector, no need to use them. The remaining LPGs found in both sectors has to be listed individually and their MPRs (Min, 25th %, median, 75th % and Max) included. Focus your comparative discussion on these drugs only. If possible, try to merge, Table 5 in to Table 4. Figures that will be listed in the revised Table 1 may show the rest.

The old Table 4 and Table 5 were merged into the new Table 5. This table has some general information on median MPRs of both OBs and LPGs in Hanam province and six areas so information on OBs was taken into account. Regarding special analyses, the median MPRs for medicines found as both product types (OBs and LPGs) can be seen in the S4 Table. The median MPRs for medicines found in both public and private sectors can be seen in the S5 Table. 

Table 5 would particularly benefit if it is changed into Figure for better general analysis. Using your reference #13 patient price cut-off point, you can point out patient prices of selected medicine. There are now standard graphics (available from HAI) in which both availability and affordability are combined in one graph. The x-axis shows availability (ranging from 0-100%, with a vertical line at 80%. The y-axis shows the affordability, with a horizontal line at the chosen cut-off point. In this graph all products have a point-position. In an ideal world all products are located in the right-lower corner of the graph, with availability at >80% and affordability below cut-off. The graph can clearly visualize where the problems are. 

(See these references for Figure https://doi.org/10.1186/s12889-021-10745-5 and https://doi.org/10.1371/journal.pone.0070836)

Thank reviewer 1 for this recommendation. What a great idea! In the Results section, we added two Figures to comprehensively analyze the availability and prices of essential medicines. However, we still use MPRs to present the y-axis (similar to the two abovementioned references). Because only 12 medicines were used to compute affordability. MPRs were twice as much data as affordability. Therefore, using MPRs to present the y-axis can supply more information.

Table 6 would particularly benefit from an extensive review. In column 2, Drug name, strength, dosage form, dose, route of administration, frequency & treatment duration, Colum 3, Treatment schedule, The total amount of drug required to cover the complete treatment regimen, Column 4, Average drug Price per Unit (USD), and Column 5, Number of day’s wage to pay for treatment. In addition, sate reasons/description why bisoprolol 5 mg cap/tab and captopril 25 mg cap/tab used for hypertension management. This briefly shows how Vietnam STG (standard treatment guideline) looks like and how closer/far to/from IPRs (cost wise).

The recommended design for Table 6 is great. However, we still want to employ the design of WHO/HAI to report affordability. The design of not only Table 6 but also other tables in our first manuscript was taken from the Survey Report Template Instructions of WHO/HAI (page 30). The Report Template can be seen in the line Chapter 12: Reporting: Supporting materials in this link: https://haiweb.org/what-we-do/price-availability-affordability/collecting-evidence-on-medicine-prices-availability

All 14 treatment courses used to compute affordability were taken from the standardized WHO/HAI Workbook (to easily compare among regions and countries). The indications of Bisoprol and Captopril were also taken from this Workbook. The original Workbook without data can be also seen in the above link. 

We also check the indications of these two medicines in the document published by the Vietnam Ministry of Health in 2018 (Vietnamese National Drug Formulary):

http://phcnhagiang.org.vn/photos/tai-lieu/duocthuquocgia20181.pdf

In this document, hypertension is one indication for both of them.

Reviewer #2:

Abstract

- Result (line 33): insert the exact p value.

The exact p-value was added (line 32). 

Introduction

Line 50: Insert reference.

Reference was added (line 50).

Line 66: ………There are several studies on medicine prices, availability and affordability conducted in Vietnam [11, 12]. Some small scale studies were carried out but not published……….

What then is the gap in knowledge that this current study aimed to fill? Does this suggest that these studies were not on Essential medicines

Some new information was added in lines 89 - 95. 

In 1985, the first National Essential Medicines List was published by the Vietnam Ministry of Health. The newest version is the 7th National Essential Medicines List released in 2018. There are several studies on medicine prices, availability and affordability conducted in Vietnam before the year 2018. From the year 2018 to now, in Vietnam, there is no study conducted to survey the availability, prices, and affordability of two types of essential medicines: originator brand and lowest-priced generic. Therefore, there is an urgent need to research essential medicines in Vietnam.

Materials and methods

Lines 86/87: ………A licensed private drugstore closest to each of the selected public facilities was selected for the private sector………

The WHO/HAI methodology also provides for survey of 'Other' pharmacies which include those domiciled in private/organization owned hospitals. Does the province have these categories of hospitals? if yes, why were they not included in the survey? if no, a brief description of the health facility set up in Hanam would suffice.

The healthcare system of Vietnam can be divided into two sectors: the public sector and the private sector. Some information on Vietnam’s healthcare system can be seen in lines 64 - 91. The Hanam healthcare system (lines 114 - 125) is similar to the national healthcare system.

Line 90: write ‘30’ in words

The number “30” was written in words (line 149).

Results:

Table 3: Urban (11 drug stores) versus rural (24 drug stores) comparison

Does this suggest that most of the districts surveyed were in the rural area? A brief sentence regarding the geography of these areas should be added in the methods section

Most of the areas in Hanam province are rural areas. The introduction to Hanam province was added in the Method section: Surveyed areas and health facilities (lines 114 - 125).

S1 Table

It is suggested that these medicines be group according to their class of drug. For example: Antibiotics, Psychotropic, Narcotics, Anti-inflammatory, Analgesics, Asthma medicines, Antihypertensive, Antidaibetic etc

The classes of surveyed medicines were added into the S3 Table - The list of surveyed medicines.

Discussion

Line 215: …….fairly high (public sector 63.2%, private sector 47.9%)……….

Recast to reflect that availability of LPGs in private sector was fairly high but that in public sector was low (47.9%)

We revised this mistake (lines 292 - 294).

Line 283: Vietnamese government has implemented many national policies to increase the availability reduce medicine prices

The authors should stress the need for the enforcement of the price regulation in the private sector. Despite that fact that prices are generally lower than those of other countries, the private sector still sells OBs higher than the LPGs to the tune of 351.34%. Unless the LPGs are of reasonable quality and efficacy in treatment of the particular disease. Are there concerns for substandard medicines in Hanam?

Some information on the prices of medicines in private medicine outlets was add in lines 386 - 390.

In private medicine outlets, regarding medicine prices, the government only requires that all medicines stocking in any medicine outlet must be labeled with prices on the medicine containers (boxes). In order to monitor and control the quality of medicines, the Department of Health in each province periodically inspects and surveys private medicine outlets. Many samples of medicines are randomly selected and tested to assess their quality.

---

## [Editor Report · Decision Letter 1]

4 Nov 2021

Availability, prices and affordability of essential medicines: a cross-sectional survey in Hanam province, Vietnam

PONE-D-21-15309R1

Dear Dr. Dinh,

We’re pleased to inform you that your manuscript has been judged scientifically suitable for publication and will be formally accepted for publication once it meets all outstanding technical requirements.

Kind regards,

Caroline Anita Lynch

Academic Editor

PLOS ONE

---

## [Editor Report · Acceptance letter]

8 Nov 2021

PONE-D-21-15309R1 

Availability, prices and affordability of essential medicines: a cross-sectional survey in Hanam province, Vietnam 

Dear Dr. Dinh:

I'm pleased to inform you that your manuscript has been deemed suitable for publication in PLOS ONE. Congratulations! Your manuscript is now with our production department. 

Kind regards, 

on behalf of

Dr. Caroline Anita Lynch 

Academic Editor

PLOS ONE